# Bioaccumulation of Metals/Metalloids and Histological and Immunohistochemical Changes in the Tissue of the European Hake, *Merluccius merluccius* (Linnaeus, 1758) (Pisces: Gadiformes: Merlucciidae), for Environmental Pollution Assessment

**Antonio Salvaggio [1], Roberta Pecoraro [2], Chiara Copat [3], Margherita Ferrante [3], Alfina Grasso [3], Elena Maria Scalisi [2], Sara Ignoto [2], Vincenza Serena Bonaccorsi [2], Giuseppina Messina [2], Bianca Maria Lombardo [2], Francesco Tiralongo [2,†] and Maria Violetta Brundo [2,*,†]**

1. Experimental Zooprophylactic Institute of Sicily "A. Mirri", 90129 Palermo, Italy; antonio.salvaggio@izssicilia.it
2. Department of Biological, Geological and Environmental Science, University of Catania, 95124 Catania, Italy; roberta.pecoraro@unict.it (R.P.); elenamaria.scalisi@unict.it (E.M.S.); saraunict@gmail.com (S.I.); quandosorridi@hotmail.it (V.S.B.); giuseppina.messina@unict.it (G.M.); bm.lombardo@unict.it (B.M.L.); francesco.tiralongo@unict.it (F.T.)
3. Department of Anatomy, Biology and Genetics, Legal medicine, Neuroscience, Diagnostic Patology, Hygiene and Public Health "G.F. Ingrassia", University of Catania, 95123 Catania, Italy; ccopat@unict.it (C.C.); marfer@unict.it (M.F.); agrasso@unict.it (A.G.)
* Correspondence: mvbrundo@unict.it
† These authors contributed equally to the manuscript.

**Abstract:** Pollution and other types of environmental stress do not spare marine environments, especially those affected by high industrial pressure. Fish, especially coastal species, are used for monitoring the marine environment because they are particularly efficient as bioindicators thanks to their ability to bioaccumulate and biomagnify along the trophic chain. The aim of this research is to evaluate the bioaccumulation and the indirect bioindication ability of the European Hake, *Merluccius merluccius* (Linnaeus, 1758), one of the most important commercial fish species of the Mediterranean Sea. Morphological and histological alterations of the main target organs, such as liver and gills, have been investigated and the results showed a steatosis in the hepatic tissue. The accumulation of heavy metals has been analyzed by inductively coupled plasma mass spectrometry and for several metals it was showed a different concentration in the two sexes. Moreover, the expression of metallothioneins 1 and Heat Shock Protein 70 has been assessed by immunohistochemistry and did not show high level of expression. We underline the importance of contamination evaluation in commercial fish species and the utilization of the ichthyofauna as bioindicator of environmental quality.

**Keywords:** commercial fish species; Mediterranean Sea; environmental health; heavy metals; biomarkers

## 1. Introduction

The significant industrial and technological development recorded in recent years has brought undoubted benefits both in terms of quality increase and life expectancy. Despite such benefits,

this rapid progress has greatly and negatively increased the impact on the environment and on human health.

An economy based on encouraging the consumption of goods and services promotes a progressive increase in anthropogenic pressure on the environment, due to the increasing loads of substances of all kinds released into the air, water and soil. Exposure to these contaminants can cause harmful effects on the health of organisms and humans.

One of the most dramatic aspects that humans will have to face himself in the immediate future is represented by waters pollution, determined by the growing release of heavy metals which, being unable to be degraded or destroyed, tend to accumulate in the environment and in organisms with consequent and substantial risks both for the various living forms and for humans who consume foods with a content of significantly high metals [1].

The effects of heavy metal pollution on the environment, on the conservation of biodiversity and on human health are now well known [1]. Immediate actions are therefore necessary both aimed at preventing the continuation of uncontrolled discharges and at imposing compliance with current regulations in terms of protection of the environment and public health. These interventions should be preceded by an environmental biomonitoring program that allows rapid identification of the risk areas.

In biomonitoring programs, the choice of species to be used is particularly important. In recent decades among the new generation bioindicators, fish have been taking on a growing interest in assessing environmental quality in various aquatic ecosystems, for their ability of bioaccumulation and biomagnification along the trophic chain [2,3]. It is also necessary to remember that fish are a fundamental component in human nutrition, representing a significant source of protein, polyunsaturated fatty acids and micronutrients. However, through the consumption of fish products, humans are thus exposed to various contaminants.

Heavy metals are elements with a characteristic shiny appearance and a good electric conductivity. In chemical reactions, heavy metals frequently occur as cations [4]. However, non-heavy elements, such as aluminum (density 1.5), can be dangerous under particular conditions. Indeed, it becomes a powerful toxic if placed in acidic waters [4].

From an environmental point of view, the term heavy metals are commonly referred to all metals or non-metals which represent a danger to health and to the environment. They are substances present in the environment as constituent elements of the same, as well as introduced following industrial emissions. They can undergo biological and chemical transformations, which entail their accumulation in the environment and in organisms, both vegetable and animal, thus manifesting their deleterious action. Although currently exposure to anthropogenic sources is of prevailing toxicological importance, exposure to natural sources has proven to be fundamental for the development, in living organisms, of detoxification mechanisms, elimination and use aimed at reducing the danger of metals. These mechanisms allow some animal species to withstand high metal concentrations, which vice versa can be toxic to others, without suffering any damage [5]. Some metals are required by organisms in limited quantities; in particular, Zn, Cu, Fe, and Mg, even if present at low concentrations, perform a series of fundamental activities for the cell behaving as essential micronutrients and participating in numerous biochemical processes responsible for cell growth and life [6].

Zinc, for example, is involved in the replication, transcription and translation processes [7,8], acts as a cofactor for over 200 metalloenzymes [9] and performs regulatory functions, as in the case of modulation of synaptic transmission [10]. Copper, in low concentrations, is essential for breathing, for defense against free radicals and for the synthesis and release of neurotransmitters [11]. Other metals, such as Cd, Cr, Al, Hg and Pb are not normally present in the cells not even in traces and therefore, even at low concentrations, they can be very toxic. These metals can cause delays in embryonic development [12,13], in growth [14–17], as well as a long series of pathologies, including cancer [18,19]. Fish species, for example, hardly eliminate the absorbed mercury and the metal halving times vary from 6 months for mussels to 2 years for pike. The accumulation in fish is greater in muscle tissue than in adipose tissue and about 90–99% of the mercury present in fish is in the form of

methylmercury, an extremely toxic form of this metal [20]. Therefore, fish in risk assessment studies are useful bioindicators and can represent an early warning system of environmental damage, which can also be used for the assessment of potential risks to human health.

*Merluccius merluccius* (Linnaeus, 1758) is a demersal fish whose depth range normally extends from 70 to 400 m, although it can be found in shallower waters and up to about 1000 m depth. Its distribution range extends from the eastern Atlantic (from Norway and Iceland to Mauritania) to the Mediterranean Sea and southern part of the Black Sea. Unlike the small specimens (<14 cm TL), which feed mainly on euphausiids and mysids, the larger specimens (>32 cm TL) of the European Hake, *M. merluccius*, are ichthyophagous [21]. Furthermore, this species has been successfully used as bioindicator [22]. Hence, considering the commercial importance of this predator, its position in the trophic web and its ability to be used as a good bioindicator of marine water pollution, our choice to use *M. merluccius* in the current study.

In this perspective, our work aims to evaluate the impact of metals/metalloids contamination in the commercial fish species *M. merluccius*, commonly known as "European Hake", sampled in the Ionian Sea. Previous research in this area of study report a total metal load higher in pelagic fish than demersal and benthic ones [23,24]. Although fish species resulted stressed by environmental conditions with a certain degree of oxidative stress in liver tissue [23], from a chemical point of view, the analyzed species were healthy for human consumption and the human risk to develop chronic systemic and carcinogenic effects due to their consumption was low [24]. Nevertheless, studies focused on the biomonitoring of the Sicilian ionic coast highlighted a significant metal load in areas subject to heavy anthropogenic pressure, particularly the Augusta coastal water, where it is hosted the largest industrial complex of Sicily [25,26].

The present study has been based on analysis of the gastrointestinal content to evaluate the presence of metals/metalloids; quantitative and qualitative analysis of metals/metalloids in the tissues; analysis of the toxicological effects of metals/metalloids through biomarkers detection (Heat Shock Proteins 70 and Metallothioneis 1) and histological analysis.

## 2. Materials and Methods

For this study, 20 specimens (10 males and 10 females) of *M. merluccius* fished in Food and Agriculture Organization of the United Nations (FAO) area 37 (Marzamemi, southeastern Sicily, Ionian Sea) were analyzed. Fish size ranged from 35 to 45 cm total length (TL), and were captured with longlines by local fishermen and transported fresh to the laboratory for analyses.

### 2.1. Metals and Metalloids Analysis

The method for extraction and quantification of metals and metalloids is described by Copat et al. [27]. Briefly, aliquots of 0.5 g of muscle, liver, gills and gastrointestinal content of fish were acid digested with 6 mL of 65% nitric acid ($HNO_3$) (Carlo Erba) and 2 mL of 30% peroxide hydrogen ($H_2O_2$-Carlo Erba) in a microwave system (Ethos Touch Control, Milestone S.r.l., Italy). Analytical determination of arsenic (As), cadmium (Cd), cobalt (Co), chromium (Cr), copper (Cu), lead (Pb), mercury (Hg), manganese (Mn), nickel (Ni), vanadium (V), selenium (Se), antimony (Sb) and zinc (Zn) was performed with an ICP-MS Elan-DRC-e (Perkin–Elmer, United States). Blanks, standard and samples were prepared with the same reagents. A multi-elements certified reference solution ICP Standard (Merck) was used for the instrument calibration. Processed blanks were used to calculate the method detection limits (MDL) based on the following equation:

$$\text{MDL} = \text{One-tailed student's } t\text{-test } (p = 0.99\%; \text{df} = n - 1) \times \text{Sr}$$

MDL (mg/kg ww) estimated for each trace elements are the following: As 0.013, Cd 0.002, Co 0.008, Cr 0.003, Cu 0.005, Pb 0.001, Hg 0.0025, Mn 0.005, Ni 0.007, V 0.025, Se 0.03, Sb 0.020 and Zn 0.109.

The quality control was performed with laboratory-fortified matrix (LFM) processed at each batch of digestion, obtaining a recovery ranges from 91.5 to 110% of the nominal concentration.

Statistical analysis was performed with the software SPSS (version 20.0, Inc., IBM, Armonk, NY, USA: IBM Corp.). Results below the MDL were elaborated as MDL/2. The normal distribution was verified using the Kolmogorov–Smirnov test. Since the low number of samples, the Mann-Whitney non-parametric test was used to compare median concentrations between tissues.

### 2.2. Histological Analysis

Histological analysis was performed according to our standard laboratory procedures [28]. Liver and gills were fixed in 4% formaldehyde (Bio-Optica, Milano, Italy) in PBS buffered to 0.1 M, pH 7.4 (Sigma Life Science) at room temperature for 48 h and processed with Tissue Processing Center TPC 15 Duo (MEDITE®, Burgdorf. Germany). The sections were stained with Haematoxylin-Eosin (HE) (Bio-Optica) and observed under optical microscope (Leica DM750, Monument, CO, USA) equipped with a digital camera (Leica DFC500, Monument, CO, USA).

### 2.3. Immunohistochemical Analysis

The immunohistochemical protocol was performed on sections to detect mouse monoclonal anti-HSP70 (Gene Tex, 1:1000) and to detect mouse polyclonal anti-MT1 (Abcam, 1:1000); secondary antibody used is FIT-conjugated goat anti-mouse IgG (Sigma-Aldrich, 1:1000). Analysis were performed according to our standard laboratory procedures [29–31]. Slides after mounted with mounting medium containing DAPI (Vectashield, Vector Laboratories, Burlingame, United States), were observed with NIKON ECLIPSE Ci fluorescence microscope and the images taken with the NIKON DS-Qi2 camera.

## 3. Results and Discussion

The chemical analysis showed that the bioaccumulation of metals in gills was higher in males for Cu ($p < 0.001$), Hg ($p < 0.01$), Se ($p < 0.001$) and Zn ($p < 0.001$), and in females for Cd ($p < 0.001$), Co ($p < 0.001$), Cr ($p < 0.01$), Pb ($p < 0.001$) and V ($p < 0.05$) (Table 1, Figure 1). Nevertheless, the concentrations of all the metals examined were low in both sexes, with higher concentrations of the essential metals Zn and Mn. The analysis of the gastrointestinal content showed a predominance content of Co ($p < 0.05$) and Mn ($p < 0.01$) in males versus an higher concentration of As ($p < 0.01$), Pb ($p < 0.001$), Ni ($p < 0.01$), V ($p < 0.01$) and Se ($p < 0.001$) in females (Table 1, Figure 2). In muscle tissue of both sex, it was observed a lower content of all metals versus the concentrations find in gill and gastrointestinal content. In females, it was found a higher concentration of Cr ($p < 0.01$), Cu ($p < 0.05$), Mn ($p < 0.001$), Ni ($p < 0.001$) and Zn ($p < 0.001$) than males. In males, only the bioaccumulation of Pb ($p < 0.001$) was found higher than females (Table 1, Figure 3).

The results obtained agreed with Bosch et al. [32] which describes a differential bioaccumulation. Particular attention is given to some heavy metals: cadmium, mercury and lead.

The authors hypothesize that the concentrations of heavy metals depend on factors influencing the absorption of metals, such as the simultaneous presence of other xenobiotics, the geographical distribution and the specific biological factors of the species. Overall, the metal concentrations found by us are comparable to those of other studies present in literature [33–35]. These concentrations of metals present in the edible portion of the fish (i.e., muscle) is acceptable as it falls within the limits of international legislation and the specimens analyzed seem to be safe for human consumption. Lead, cadmium and mercury were recorded with values well below the limits imposed by law (0.30 mg/kg for lead, 0.05 mg/kg for cadmium, 0.50 mg/kg for mercury).

**Table 1.** Descriptive statistics of metals and metalloids (mg/kg ww) in *Merluccius merluccius*.

| Tissue | Sex | Statistics | As | Cd | Co | Cr | Cu | Pb | Hg | Mn | Ni | V | Se | Sb | Zn |
|---|---|---|---|---|---|---|---|---|---|---|---|---|---|---|---|
| Gill | Males | Min. | 1.115 | 0.005 | 0.465 | 2.468 | 4.746 | 2.459 | 0.030 | 78.25 | 1.452 | 4.598 | 0.246 | 0.021 | 16.43 |
| | | Max. | 1.924 | 0.010 | 0.513 | 3.215 | 5.483 | 5.896 | 0.058 | 130.5 | 3.145 | 9.452 | 0.495 | 0.059 | 19.75 |
| | | Mean | 1.536 | 0.007 | 0.496 | 2.851 | 5.137 | 3.706 | 0.043 | 104.7 | 2.408 | 6.501 | 0.376 | 0.034 | 18.76 |
| | | S.D. | 0.234 | 0.002 | 0.017 | 0.183 | 0.260 | 1.032 | 0.009 | 18.12 | 0.559 | 1.400 | 0.074 | 0.012 | 1.094 |
| | Females | Min. | 1.248 | 0.022 | 0.745 | 2.853 | 3.145 | 5.424 | 0.021 | 78.25 | 2.139 | 6.300 | 0.156 | 0.021 | 8.120 |
| | | Max. | 1.570 | 0.037 | 0.985 | 3.851 | 4.926 | 7.952 | 0.040 | 120.3 | 2.770 | 9.485 | 0.235 | 0.062 | 12.37 |
| | | Mean | 1.429 | 0.029 | 0.880 | 3.354 | 4.163 | 6.486 | 0.032 | 97.80 | 2.517 | 8.045 | 0.189 | 0.044 | 10.11 |
| | | S.D. | 0.106 | 0.005 | 0.079 | 0.306 | 0.483 | 0.735 | 0.006 | 11.53 | 0.200 | 1.253 | 0.024 | 0.013 | 1.394 |
| Gastrointestnal content | Males | Min. | 0.302 | 0.001 | 0.034 | 0.345 | 1.952 | 0.135 | 0.174 | 1.324 | 0.041 | 0.180 | 0.141 | <0.020 | 8.125 |
| | | Max. | 0.601 | 0.007 | 0.064 | 0.699 | 2.770 | 0.195 | 0.251 | 1.520 | 0.095 | 0.264 | 0.195 | <0.020 | 12.45 |
| | | Mean | 0.405 | 0.003 | 0.048 | 0.544 | 2.358 | 0.165 | 0.197 | 1.390 | 0.066 | 0.220 | 0.177 | <0.020 | 9.873 |
| | | S.D. | 0.084 | 0.002 | 0.009 | 0.107 | 0.270 | 0.020 | 0.023 | 0.053 | 0.018 | 0.028 | 0.016 | / | 1.122 |
| | Females | Min. | 0.421 | 0.001 | 0.021 | 0.485 | 1.214 | 0.214 | 0.182 | 1.100 | 0.065 | 0.215 | 0.234 | <0.020 | 8.420 |
| | | Max. | 0.621 | 0.009 | 0.064 | 0.741 | 3.254 | 0.512 | 0.260 | 2.164 | 0.164 | 0.354 | 0.485 | <0.020 | 12.48 |
| | | Mean | 0.505 | 0.005 | 0.037 | 0.620 | 2.092 | 0.324 | 0.210 | 1.903 | 0.099 | 0.280 | 0.357 | <0.020 | 10.88 |
| | | S.D. | 0.067 | 0.003 | 0.011 | 0.095 | 0.585 | 0.088 | 0.027 | 0.348 | 0.029 | 0.049 | 0.087 | / | 1.331 |
| Muscle | Males | Min. | 0.111 | 0.001 | <0.008 | 0.325 | 0.218 | 0.007 | 0.005 | 0.230 | <0.007 | <0.025 | 0.075 | <0.020 | 0.796 |
| | | Max. | 0.194 | 0.009 | <0.008 | 0.852 | 0.324 | 0.021 | 0.016 | 0.465 | <0.007 | <0.025 | 0.164 | <0.020 | 1.500 |
| | | Mean | 0.151 | 0.004 | <0.008 | 0.522 | 0.272 | 0.014 | 0.009 | 0.363 | <0.007 | <0.025 | 0.111 | <0.020 | 1.101 |
| | | S.D. | 0.033 | 0.002 | / | 0.146 | 0.041 | 0.005 | 0.003 | 0.072 | / | / | 0.023 | / | 0.213 |
| | Females | Min. | 0.114 | 0.002 | <0.008 | 0.596 | 0.222 | 0.001 | 0.005 | 1.108 | 0.011 | <0.025 | 0.085 | <0.020 | 1.745 |
| | | Max. | 0.384 | 0.006 | <0.008 | 0.771 | 0.513 | 0.009 | 0.014 | 1.345 | 0.035 | <0.025 | 0.254 | <0.020 | 2.224 |
| | | Mean | 0.212 | 0.004 | <0.008 | 0.692 | 0.341 | 0.003 | 0.010 | 1.201 | 0.021 | <0.025 | 0.137 | <0.020 | 2.072 |
| | | S.D. | 0.086 | 0.001 | / | 0.055 | 0.079 | 0.002 | 0.003 | 0.078 | 0.008 | / | 0.057 | / | 0.141 |

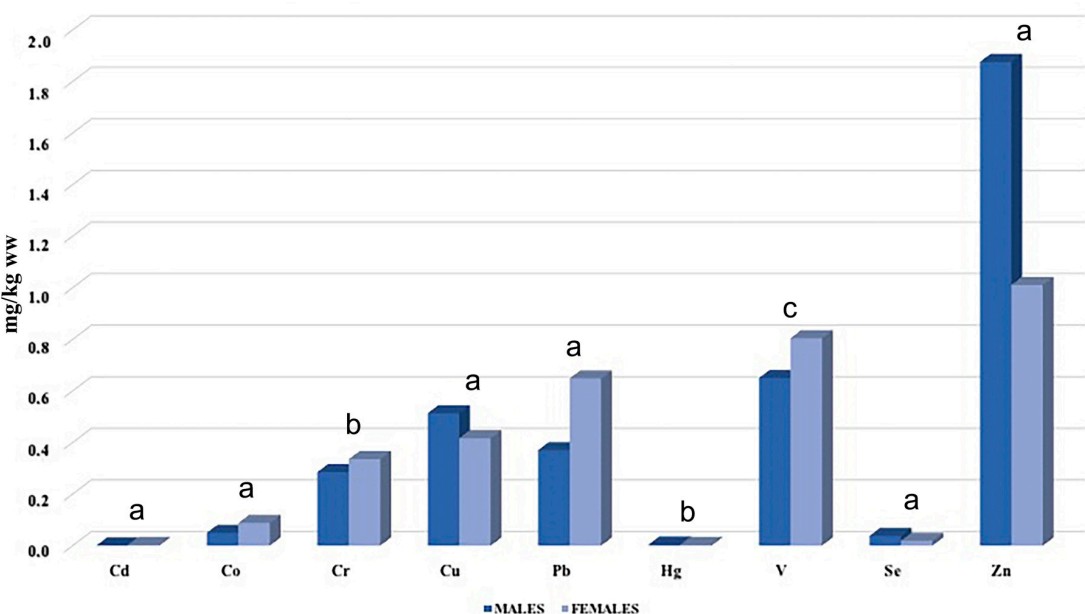

**Figure 1.** Mean values of metals and metalloids concentrations in gills (mg/kg ww). Significant differences between sex: a, $p < 0.001$; b, $p < 0.01$, c: $p < 0.05$.

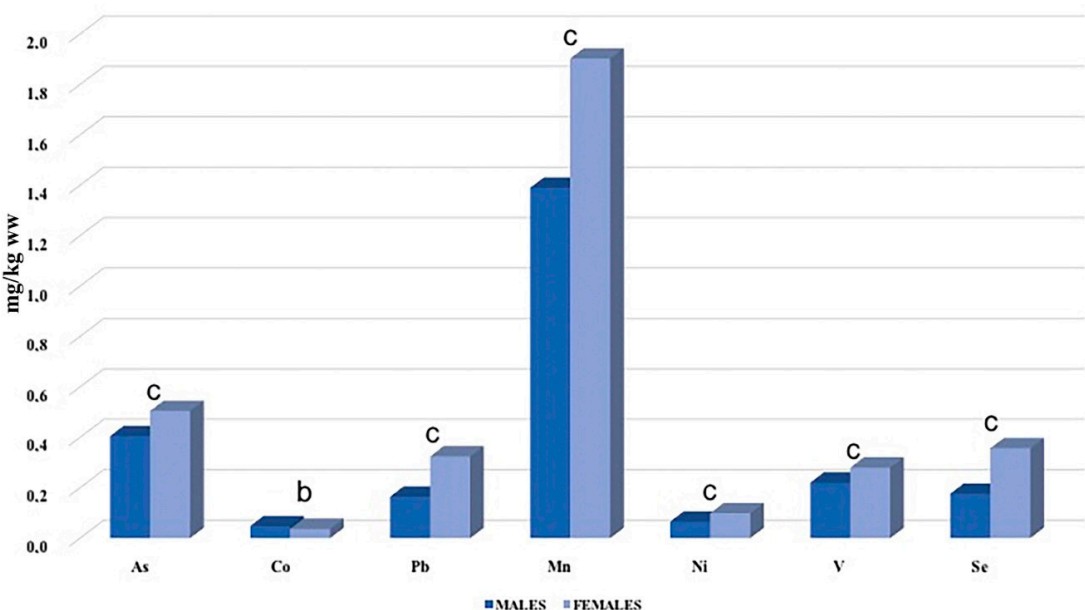

**Figure 2.** Mean values of metals and metalloids concentrations in gastrointestinal content (mg/kg ww). Significant differences between sex: b, $p < 0.01$, c: $p < 0.05$.

As far as histological investigations are concerned, morphological anomalies of the gill lamellae were not found in the analyzed specimens (Figure 4A,B), but marked liver steatosis was evident in the liver sections (Figure 4C,D). The immunohistochemical investigation revealed a weak expression of MT1 (Figure 5A,C) and HSP70 (Figure 5B,D) in both the target organs (gills and liver) of both sexes.

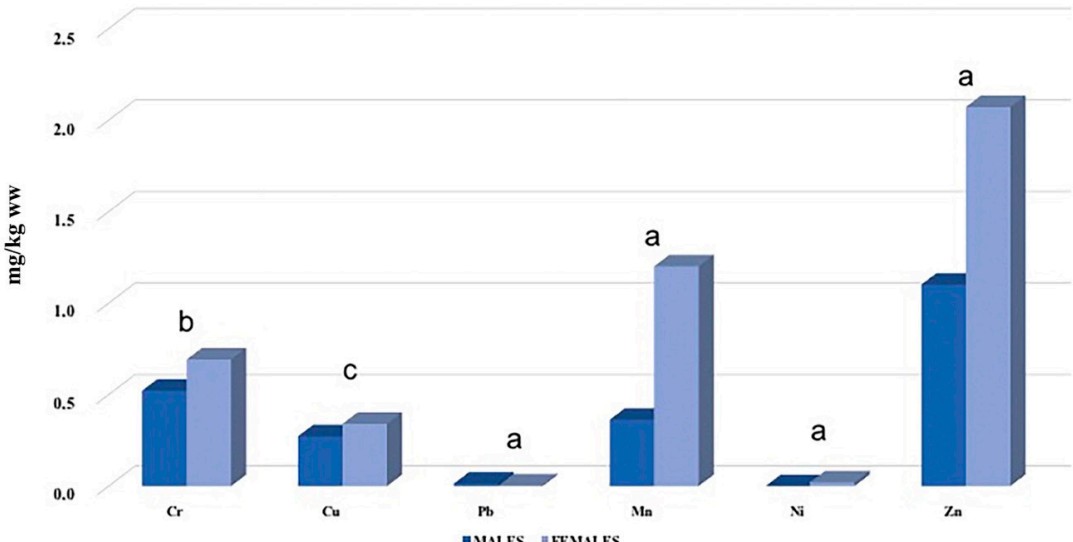

**Figure 3.** Mean values of metals and metalloids concentrations in muscle (mg/kg ww). Significant differences between sex: a, $p < 0.001$; b, $p < 0.01$, c: $p < 0.05$.

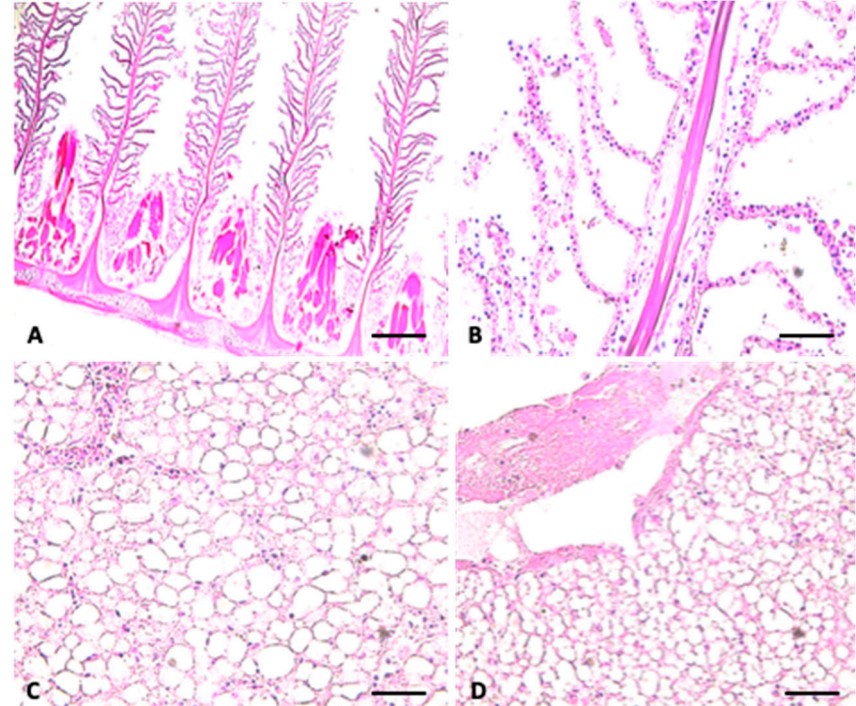

**Figure 4.** Sections stained with Hematoxylin-Eosin. (**A**,**B**); gills sections (**C**,**D**); male liver sections, is evident a diffuse steatosis. Scale bar A,C and D: 200 μm; B 100 μm.

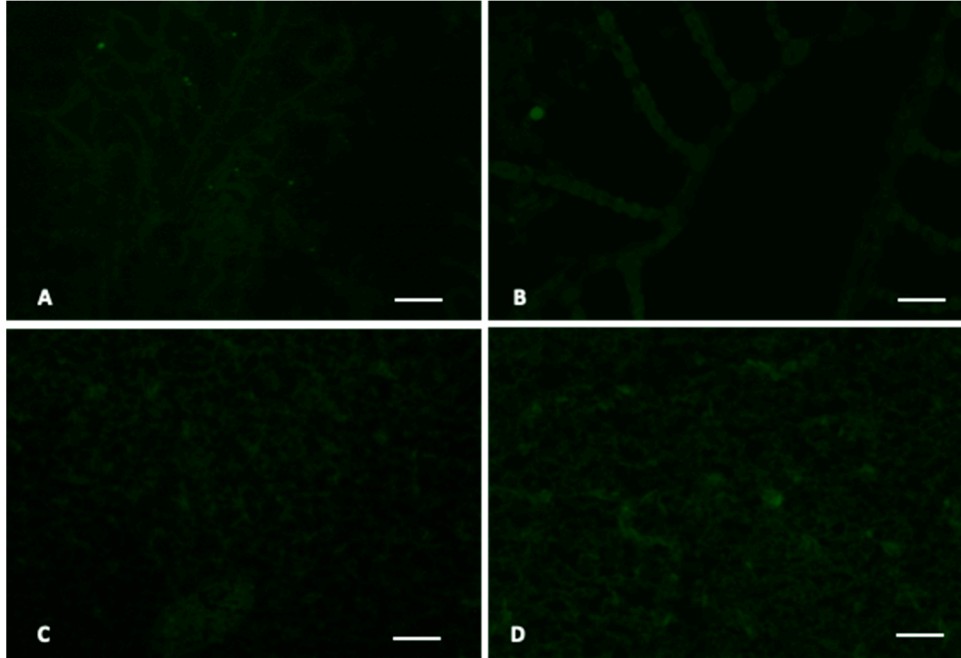

**Figure 5.** MT 1 expression in male gills (**A**) and liver (**C**). HSP70 expression in male gills (**B**) and in liver (**D**). Using specific antibodies anti-MT1 and anti-HSP70 it has been showed a weak expression of these proteins in cells (green). Scale bar A, C and D: 200 μm; B: 50 μm.

The histological analysis, despite being non-specific about pathogenic noxae, offers the possibility of highlighting any alterations in the gill and liver tissues. The use of these tissue is an efficient biomarker in a preliminary screening for fish and water quality. *Merluccius merluccius*, being a voracious predator, can be considered a useful species for monitoring the environmental situation over time through bioaccumulation and biomagnification studies. Heavy metals contamination in fishery products remains an almost inevitable condition due to the environmental situation facing most of the national fishing areas. Therefore, as already pointed out by European Food Safety Authority (EFSA) [36] in order to reduce this risk to an acceptable level, it is necessary to bridge the lack of information on the conscious consumption of the various fish species and, more generally, it is necessary that consumers can receive a concrete information on the possible risks associated with the consumption of certain species.

In recent decades, with the rapid evolution of molecular, biochemical and pathological technologies, biomarkers have found a very wide application. The toxicity of a contaminant on an organism is usually expressed at a biochemical and molecular level and, because of this, at cellular and tissue levels. As a response to the action of a toxic agent, the body develops adaptive responses that tend to bring the system back into a balanced state. However, when the homeostatic mechanism is not sufficient to balance the action of the toxic agent, the negative effect is manifested at the cellular, tissue and eventually organ level. It can therefore be said that the different responses, homeostatic and otherwise, generated by an organism towards a toxic agent, represent potential markers that can be used in ecotoxicological investigations [37,38].

It can be said that the gills, because of the role of primary importance in interfacing the fish with the surrounding aquatic environment and for their demonstrated reactivity and sensitivity, are optimal candidates for the role of versatile biomarkers in the biomonitoring of aquatic ecosystems. They lend themselves to a study in the field of environmental protection, also aimed at protecting the human being as a user, direct and indirect, of the resources that natural ecosystems can offer. However, also the investigation of ecological indices represents a good method for the monitoring of the environmental

quality [39] and these types of studies should be associated to the bioaccumulation studies for a more complete and wide point of view.

## 4. Conclusions

The diversity of cultures bordering the Mediterranean and the level of anthropization present make interventions aimed at environmental protection difficult. The same international fishing legislation of the Mediterranean Sea is applied with different weights, according to the different social and economic realities on which it acts. To protect the Mediterranean environment, which currently has different pressures at the expense of coastal and marine habitats, it is increasingly necessary, in addition to implementing and applying existing environmental laws, to also adopt integrated approaches based on knowledge of ecosystems in order to better understand their organization and functioning. The result of these negative impacts is an ecosystem in which the survival of fish species is dependent on human intervention. This study analyzes some of the complex variables that currently affect the fish populations of the Mediterranean Sea, intended both as a fish resource of great commercial value, and as a symbol of an ecosystem that now lives below the limits of sustainability. In particular, the contamination level recorded by us in *M. merluccius* in the current study suggested that this important commercial species can be considered sure for human consumption, although we recorded a marked liver steatosis that can be the result of stressing environmental conditions. Further investigation in this direction may help to better understand contamination dynamics and effects on fish population and their potential risks to human health.\

**Author Contributions:** All authors have made substantial contributions to the conception and the design of the work; have approved the submitted version have agree to be personally accountable for the author's own contributions and for ensuring that questions related to the accuracy or integrity of any part of the work, even ones in which the author was not personally involved, are appropriately investigated, resolved, and documented in the literature. Methodology, A.S., V.S.B., C.C., A.G.; visualization, S.I., G.M., M.F., B.M.L.; writing, review and editing, M.V.B., F.T., R.P., E.M.S., A.S.; conceptualization M.V.B. and F.T. All authors have read and agreed to the published version of the manuscript.

**Funding:** This study was funded by the Department of Biological, Geological and Environmental Science and by the Department of Medical Sciences, Surgical and Advanced Technologies "G.F. Ingrassia"—Hygiene and Public Health, University of Catania.

**Conflicts of Interest:** The authors declare no conflict of interest.

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
