# Peer review of "Bioaccumulation of Metals/Metalloids and Histological and Immunohistochemical Changes in the Tissue of the European Hake, Merluccius merluccius (Linnaeus, 1758) (Pisces: Gadiformes: Merlucciidae), for Environmental Pollution Assessment"

_jmse, doi:10.3390/jmse8090712_

Round 1

Reviewer 1 Report

I am presenting the remarks on the manuscript JMSE 903605. First, I suggest to change the title of the manuscript.  In fact,  the authors compared the concentration of the metals and histological, immunohistochemical changes in the tissue of female and male of the European hake.

The discussion of the environmental risk assessment should be done in relation to level of metal contamination in areas, which were the source of the samples. I understand that the intention of the authors is present a general quality of the food fish collected in Mediterranean Sea. But, anyway, the authors did not present any specific information about the place of the fish origin, which is important to be sure that the collection of fish was representative for marine environment contamination - fish area 37 is quite huge, with differently contaminated sites. It is important, because for such study the authors used only ten male and ten female specimens.

In the Introduction part, the authors should present also some information on typical habitat or food of the European hyke to convince that this species is good representative as bioindicator. Possible a more elaborated information of this species using as common edible fish is needed.

An examples to consider: Carrozzi et al., 2019. Regional studies in Marine Sciences. Martinez-Morcillo et al., 2019. Science of Total Environment; Philips,2012 and 2014. The Egyptian Journal of Aquatic Research.

Histograms need more specific description. The authors wrote about Mann-Whitney test to compare concentration of metals in different fish tissues, so in the figures should be presented calculated statistically significant differences. It should be also presented differences between sexes.

I do not know why the authors presented Results and Discussion and separately Discussion.

Author Response

All required information and corrections have been inserted into the text. Please see the attachment.

Reviewer 2 Report

The article is devoted to the important aspects of identifying the influence of heavy metals on human health since the object of the study is one of the most popular fish species caught in the FAO 37 zone. The methods used are adequate to the tasks set, the results are clear and well discussed in the concluding part of the manuscript. The work on maintenance is among the similar articles important for assessing the quality of commercial fish, supplementing the previously obtained data with new conclusions. It is particularly interesting that this study allows one to determine the current state of the aquatic ecosystem where fishing is carried out. The article can be published in the journal with minor corrections indicated in the attached file. The comments mostly concern the technical side of the manuscript design, especially the presented figures. Their quality should be improved so that the reader can understand the conclusions of the authors. English is of satisfactory quality but could be improved in some places.

Author Response

All required changes have been made.Please see the attachment.

Round 2

Reviewer 1 Report

The article has much better quality, but I have still three remarks. 

The authors did not present the situation in the Ionian sea regarding heavy metals pollution.    In my opinion it would be much better, when they will refer it to more specific situation from the area, where the fish were caught. The examples: Copat et al., 2012. Heavy metal concentrations in fish from Sicily (Mediterraean sea) and evaluation of possible health risk to consumers. Bull Environ Contam Toxicol; Caruso 2011 Response of benthic foraminifera to heavy metal contamination in marine sediments (Sicilian coasts, Mediterranean Sea). Chem Ecol

The next remark - the figures/tables descriptions should be more informative  - for example what do mean yellow or green stripes in Tables 1. In figs. 1-3,. letters above bars should be described as indicating significant differences between sexes, I suppose

The authors presented excessive general conclusion text. For example when  the authors wrote: "The most important aspects to consider are: 1) the pollution produced by urbanization and industrial activities; 2) the sustainability of the exploitation of fishing and aquaculture resources;  3) the presence of inadequate information." the authors did not present any specific cases referred to the text above and connected with the results of this study

Author Response

Reply Referee 1 Round 2

The article has much better quality, but I have still three remarks. 

The authors did not present the situation in the Ionian sea regarding heavy metals pollution.  In my opinion it would be much better, when they will refer it to more specific situation from the area, where the fish were caught. The examples: Copat et al., 2012. Heavy metal concentrations in fish from Sicily (Mediterraean sea) and evaluation of possible health risk to consumers. Bull Environ Contam Toxicol; Caruso 2011 Response of benthic foraminifera to heavy metal contamination in marine sediments (Sicilian coasts, Mediterranean Sea). Chem Ecol

Thanks for your suggestions. We have inserted specific bibliography and expanded the information on specific situation from the area where the fish were caught.

The next remark - the figures/tables descriptions should be more informative  - for example what do mean yellow or green stripes in Tables 1. In figs. 1-3,. letters above bars should be described as indicating significant differences between sexes, I suppose

Done. Yellow and green stripes in Tables 1 were left in error (delete).

The authors presented excessive general conclusion text. For example when  the authors wrote: "The most important aspects to consider are: 1) the pollution produced by urbanization and industrial activities; 2) the sustainability of the exploitation of fishing and aquaculture resources;  3) the presence of inadequate information." the authors did not present any specific cases referred to the text above and connected with the results of this study

Done. The conclusion have been changed as required.
